 

# Health seeking behaviours and private sector delivery of care for non-communicable diseases in low- and middle-income countries: protocol for a systematic review

Callum Brindley ,[1,2] Nilmini Wijemunige ,[1,3] Charlotte Dieteren ,[1,2] Judith Bom ,[1,2] Maarten Engel ,[4] Bruno Meessen ,[5] Igna Bonfrer [1,2]

¹Erasmus School of Health Policy and Management, Erasmus University Rotterdam, Rotterdam, The Netherlands
²Erasmus Centre for Health Economics, Erasmus University Rotterdam, Rotterdam, The Netherlands
³Institute for Health Policy, Colombo, Sri Lanka
⁴Erasmus Medical Centre, Erasmus University Rotterdam, Rotterdam, The Netherlands
⁵World Health Organization, Geneva, Switzerland

**Correspondence to**
Callum Brindley;
brindley@eshpm.eur.nl

## ABSTRACT

**Introduction** The burden of non-communicable diseases (NCDs) has increased substantially in low- and middle-income countries (LMICs), and adapting health service delivery models to address this remains a challenge. Many patients with NCD seek private care at different points in their encounters with the health system, but the determinants and outcomes of these choices are insufficiently understood. The proposed systematic review will help inform the governance of mixed health systems towards achieving the goal of universal health coverage. This protocol details our intended methodological and analytical approaches, based on the Preferred Reporting Items for Systematic Reviews and Meta-Analyses (PRISMA).

**Methods and analysis** Following the PRISMA approach, this systematic review will develop a descriptive synthesis of the determinants and outcomes of private healthcare utilisation for NCDs in LMICs. The databases Embase, Medline, Web of Science Core Collection, EconLit, Global Index Medicus and Google Scholar will be searched for relevant studies published in English between period 1 January 2010 and 30 June 2022 with additional searching of reference lists. The study selection process will involve a title-abstract and full-text review, guided by clearly defined inclusion and exclusion criteria. A quality and risk of bias assessment will be done for each study using the Mixed Methods Appraisal Tool.

**Ethics and dissemination** Ethical approval is not required because this review is based on data collected from publicly available materials. The results will be published in a peer-reviewed journal and presented at related scientific events.

**PROSPERO registration number** CRD42022340059

## STRENGTHS AND LIMITATIONS OF THIS STUDY

⇒ This is a systematic review protocol that proposes a timely contribution to the challenge of adapting healthcare service delivery to prevent, treat and manage non-communicable diseases in low- and middle-income countries (LMICs).
⇒ We define the private sector broadly to capture its breadth and complexity.
⇒ Although we will consider all LMICs, this systematic review is limited to studies published in English, so does not include relevant literature in other languages.
⇒ We limit our attention to literature from the past decade which we contend is most relevant given recent changes in disease burden, risk factors and demographics in LMICs.
⇒ We will not perform a meta-analysis because we expect highly diverse study characteristics, including design type, setting, intervention and outcome.

## INTRODUCTION

The rising burden of non-communicable diseases (NCDs) in low- and middle-income countries (LMICs)[1 2] calls for policy responses that engage the whole health system, including private providers. While the appropriate role of the private sector in the delivery of healthcare has been heavily debated,[3–6] it is well known that patients seek private care at different points in their encounters with the health system.[7–10] Research into the reasons for visiting private providers and the comparative quality of this care in LMICs has largely focused on infectious diseases, and maternal and child health.[11]

With regard to NCDs, the determinants and outcomes of private healthcare utilisation in LMICs remain insufficiently understood. Importantly, NCDs have multiple aetiologies (including behavioural, environmental and genetic) that require an organisation of services and delivery of care that is different from that of acute illnesses.[12] Specifically, a focus on prevention and primary care that empowers patients and prioritises information sharing and coordination across levels of care and providers.[12 13] This is imperative for the achievement of Sustainable Development Goal (SDG) target 3.4—a

one-third reduction in premature mortality from NCDs by 2030.

Adapting health service delivery models to respond to NCDs remains a challenge.[14] In many LMICs, national NCD strategies tend to focus on the public sector to the exclusion of the private sector despite the two being entwined elements of the whole health system.[15–17] Many LMICs with universal health coverage (UHC) benefits packages also only offer limited coverage for NCDs, such that patients must pay out-of-pocket not only in private health facilities but even in public ones.[18] The risk is that this discourages and distorts health seeking behaviours, leading to ineffective care, poor health outcomes and catastrophic health spending.[18 19] Understanding better the reality of health seeking behaviours for NCDs, including the possible role taken by private providers, will be key for the design and implementation of appropriate strategies and interventions, across the different building blocks of the health system.[20]

This study aims to synthesise the scattered evidence on how people with NCDs choose their healthcare providers in LMICs, and the outcomes of these choices, in the understudied private sector. This will provide important new insights into the determinants and outcomes of private healthcare utilisation for NCDs in LMICs. In particular, we concentrate on the contextual and individual factors that influence provider choice, patterns of utilisation, quality of care and financial protection. We use the notion of the patient journey and draw on existing conceptual frameworks of health seeking behaviour and the facilitators and barriers to accessing healthcare.[21–25] This systematic review has been initiated by the WHO's Department of Noncommunicable Disease and Department of Health System Financing and Governance. It follows work by a WHO Advisory Group on Governance of the Private Sector and a systematic review of private sector delivery of quality care for maternal, newborn and child health in LMICs.[26 27] This review will contribute to an understanding of health seeking behaviour for NCDs in the private sector in LMICs, which will in turn help inform the governance of mixed health systems in the pursuit of UHC and the achievement of the SDGs.

## METHODS AND ANALYSIS
### Aim and research questions
This systematic review on the determinants and outcomes of private healthcare utilisation is built around the questions— *What factors influence private healthcare seeking behaviour for individuals with NCDs in LMICs? From whom is healthcare obtained, what are the patterns of utilisation, what are the determinants that influence the use of private as opposed to public healthcare, and what are the outcomes of this healthcare seeking?* This review allows us to determine the size and nature of the current literature and to identify major knowledge gaps that relate to mixed healthcare systems with entwined public and private providers of NCD care in LMICs.

**Table 1** PICOTS criteria used in the systematic review

| | |
|---|---|
| Population | Adults aged 18 years or older and households at risk, or diagnosed with at least one of the following non-communicable diseases: cardiovascular diseases, diabetes, chronic respiratory diseases or cancers (tracheal, bronchus and lung; colon and rectum; pancreas; stomach; breast and prostate) |
| Interventions | Private sector provision of healthcare services all levels (eg, primary health clinic to hospital) that involves the prevention, diagnosis, treatment and/or management of non-communicable disease |
| Control | Not applicable |
| Outcomes | Determinants and outcomes of healthcare seeking behaviour, at the individual and population level |
| Timeframe | 1 January 2010 to 30 June 2022 |
| Setting | Low- and middle-income countries following the World Bank 2022 classification[34] |

### Study design and protocol
This study is guided by the Preferred Reporting Items for Systematic Reviews and Meta-Analyses (PRISMA) and its extension for literature searches PRISMA-S.[28 29] PRISMA is a systematic approach to map and synthesise existing evidence, identify knowledge gaps and inform future research. This review was registered with PROSPERO on 15 June 2022. We checked that there were no current or in-progress systematic reviews on the same topic by searching PROSPERO, the Research Registry, and the Open Science Framework.

### Inclusion/exclusion criteria
Table 1 outlines the Populations, Interventions, Control, Outcomes, Timeframe, Setting (PICOTS) criteria to be used. We focus on adults aged 18 years or older and define the scope of healthcare to encompass the prevention, diagnosis, treatment and/or management of disease at all level (eg, primary health clinic to hospital). We restrict our search to four groups of NCDs responsible for over 80% of all premature NCD deaths and identified by SDG target 3.4, specifically cardiovascular diseases, cancers, respiratory diseases and diabetes. With regard to cancers, we limit our scope to the top five cancers with the greatest disease burden for each sex.[30] We use the WHO's operational definition of the private health sector as the individuals and organisations that are neither owned nor directly controlled by governments, and are involved in the provision of health services (ie, formal and informal providers as well as for-profit and non-profit entities).[31] Our outcomes of interest are the determinants and outcomes of health seeking behaviour at both the individual and collective level, which are described in more detail under the search elements and key terms. We draw on existing theoretical approaches to health seeking behaviour and the facilitators and barriers to accessing healthcare.[21–25] Figure 1 depicts our conceptual framework of the non-linear interaction of the determinants and outcomes along the patient journey of seeking private healthcare for NCDs. We focus on the timeframe

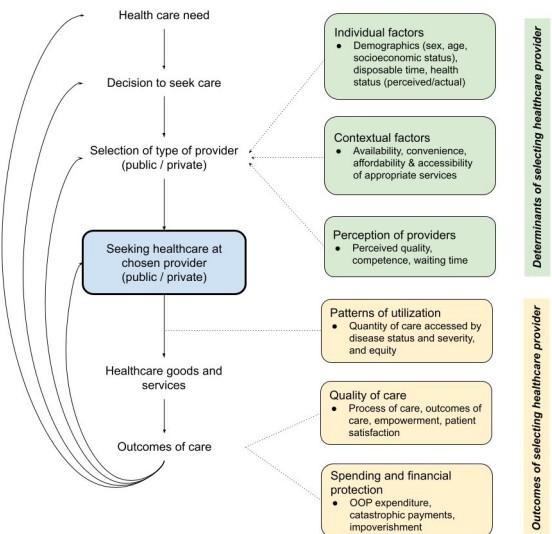

**Figure 1** Conceptual framework.

**Table 2** Search elements and key terms

| Search element | Key terms |
|---|---|
| What: non-communicable diseases | non-communicable diseases; cardiovascular disease; heart attack; stroke; myocardial infarction; hypertension; diabetes; cancer (top five for each sex: lung, breast, colorectum, prostate, stomach, liver, cervix, thyroid); chronic respiratory disease; chronic obstructive pulmonary disease; asthma) |
| Where: healthcare providers in low- and middle-income countries | hospital, health clinic, community health centre, health centre, health post, family practitioner, general practitioner, primary care, secondary care, tertiary care, specialist, home care, nursing home, pharmacy, public provider, private provider, nonprofit provider, faith-based provider, low-income countries, middle-income countries, lower-middle income countries, upper-middle income countries, list of all LMICs |
| Why and what happens: determinants and outcomes of healthcare decisions/ health seeking behaviour | quality; cost; affordability; accessibility; proximity; distance; opening hours; in-network provider; reimbursement; reputation; health service delivery; health outcomes; utilisation; out-of-pocket payment; indirect costs; catastrophic healthcare expenditure; debt; health inequalities; health information; supply; fee for service; empowerment; patient reported experience |

from January 2010 to June 2022, which is defined by large increases in the NCD burden in LMICs and the first High-Level Meeting of the UN General Assembly on the Prevention and Control of Non-Communicable Diseases.[30 32] This also avoids duplication of a broader systematic review into the private provision of health services in LMICs conducted in 2011.[33] Lastly, the study looks at settings in LMICs as defined by the World Bank classification for 2022 (ie, countries with a gross national income per capita of \$4095 or less).[34]

For inclusion, studies must adhere to all elements defined in table 1. They must be in English and published in a peer-reviewed journal. We will include both qualitative and quantitative studies, including randomised and non-randomised study designs (ie, cross-sectional, case-control and cohort observational studies). Editorials, commentaries, reviews and protocols will be excluded.

Studies must address private sector healthcare and may do so in a comparison with the public sector. However, we will exclude studies that only report aggregated data (ie, private and public health sector outcome data combined). Lastly, we will also exclude vaccination programmes associated with NCDs (eg, human papillomavirus) because they are government-led, as well as settings with irregular contextual circumstances (eg, conflict, economic and political crises) since they are less likely to offer generalisable insights.

## Search strategy

Our search strategy was developed with the assistance of an information specialist (ME). The following electronic databases will be searched: Embase, Medline, Web of Science, EconLit and Global Index Medicus.[35] We will also search Google Scholar and download the 200 most relevant references in line with common practice.[36] Table 2 provides the search elements, and key terms we will use. Specifically, it contains terms for (1) NCDs, (2) healthcare providers in LMICs and (3) the determinants and outcomes of health seeking behaviour and related terms. The search was

developed in Embase, optimised for sensitivity, then translated to other databases following the method described by Bramer et al.[37] The search strategies for Embase and Medline use relevant thesaurus terms from Emtree and Medical Subject Headings, respectively. In all databases, the terms will be searched in titles, author keywords and major index terms; the search element for setting (healthcare providers in LMICs) will also be searched in abstracts. The full search strategies of all databases are provided in online supplemental material. References will be imported into EndNote, and duplicates removed.[38] Reference lists of relevant reviews and all included articles will also be screened for potential additional sources missed by the search. We do not plan to contact authors or subject expects.

## Screening process

The articles resulting from the search will be screened by a team of two pairs (CB-JB and NW-CD) working in the Rayyan Reference Manager.[39 40] After removing duplicates, each article will be title-abstract screened independently by two individuals using the eligibility criteria described earlier. To reduce bias, author names will not be visible during this screening stage. Before commencing screening by pairs, we will also conduct a pilot phase in which a random selection of 100 articles will be reviewed and discussed jointly to harmonise our approach. Articles adhering to all criteria will be selected for full-text review. Title-abstracts will be screened in three lots. Any conflicts will be reviewed within the pair at the conclusion of reviewing each lot, to allow for further

harmonisation within the pair. If an inclusion/exclusion decision cannot be made based on the title and abstract, the article will be advanced to full-text screening. In the case of a conflicting inclusion/exclusion decision, the other pair in the team will also review the article, and the supervising researcher (IB) will provide a final adjudication if needed. Articles selected for full-text screening will be reviewed by two individuals, and a decision to exclude at this stage will be documented. Confirmed texts for inclusion in the systematic review will be advanced to the next stage of data extraction.

### Data extraction

We will pilot and refine a data extraction framework including *study details* such as authors, title and year; and *study characteristics* such as research design, sample size, studied country, disease and provider type. We will extract the *determinants of selecting a healthcare provider*, grouped into three categories: (1) *individual factors* such as demographics and health; (2) *contextual factors* such as availability and accessibility of healthcare and (3) *perception of providers* such as quality and competence. Lastly, we extract *outcomes of selecting a healthcare provider*, which we will group into three categories: (1) *patterns of utilisation* including characteristics of patients and equity issues; (2) *quality of care* including the Donabedian model[25] domains of process and outcomes as well as patient satisfaction and empowerment and (3) *spending and financial protection* as captured by indicators such as catastrophic health spending and impoverishment. We will also consider sociocultural, economic and political contextual factors, which are important determinants of health seeking in LMICs. Each full-text article included in the systematic review will be read by two individuals with one person taking the lead to extract the data, and the second person reviewing this work.

In the Quality assessment section, we explain how we will use the Mixed Methods Appraisal Tool (MMAT) to assess the methodological quality

### Quality assessment

We will use the MMAT to assess the methodological quality and risk of bias of the reviewed full-text articles.[41] MMAT is a critical appraisal tool intended for systematic reviews that include diverse study designs (ie, quantitative, qualitative and mixed methods). Each full-text article will be assessed by a team of two pairs (CB-JB and NW-CD) with one person taking the lead and the second person reviewing for completeness and accuracy. Discrepancies and disagreements will be resolved through discussion.

### Synthesis

The team will meet to discuss and analyse the data extracted, which we will consolidate using a descriptive synthesis including a summary of the evidence, gaps and limitations. As described earlier in the data extraction, we will synthesise the results into three categories of determinants of selecting a healthcare provider (individual factors, contextual factors and perception of providers) and three categories of outcomes of selecting a healthcare provider (patterns of utilisation, quality of care and

spending and financial protection). We will also develop a synthesis of the results that concern the interaction of health seeking determinants and outcomes. A flow diagram of the inclusion/exclusion pathways and the descriptive statistics of the included studies and their outcomes will be developed to complement the analysis. We will not perform a meta-analysis because we expect highly diverse study characteristics, including design type, setting, intervention and outcome.

### Patient and public involvement

None

## ETHICS AND DISSEMINATION

Ethical approval is not required because this is a review and collection of data based on publicly available materials. The results will be presented at events attended by policy makers, academics and healthcare practitioners from LMICs, and later published in a topic-relevant journal. Understanding the reality of health seeking behaviours for NCDs will be key for the design and implementation of interventions to strengthen the building blocks of the health system: from regulation of the health workforce and provider payment reform to benefit package design and the digitalisation of patient files. This systematic review will be a timely contribution to the challenge of adapting healthcare service delivery to prevent, treat and manage NCDs in LMICs.

**Acknowledgements** We wish to acknowledge the valuable contributions of Annalise Belloni, Owen O'Donnell, Tom van Ourti and Eddy van Doorslaer who provided input for this systematic review protocol.

**Contributors** IB, BM, ME and CB conceptualised the protocol. CB wrote the first draft with input from NW, CD, JB, ME and IB. All authors contributed to subsequent revisions and approved the protocol prior to its submission. All authors are accountable for the research presented.

**Funding** This work was supported jointly by the Non-Communicable Disease Department and Health System Financing and Governance Department of the World Health Organization (Grant number 202758205). The content of this study is the sole responsibility of the authors and does not represent the official views of World Health Organization.

**Disclaimer** The author is a staff member of the World Health Organization. The author alone is responsible for the views expressed in this publication and they do not necessarily represent the views, decisions or policies of the World Health Organization.

**Competing interests** None declared.

**Patient and public involvement** Patients and/or the public were not involved in the design, or conduct, or reporting, or dissemination plans of this research.

**Patient consent for publication** Not applicable.

**Provenance and peer review** Not commissioned; externally peer reviewed.

**ORCID iDs**
Callum Brindley http://orcid.org/0000-0002-5478-3974
Nilmini Wijemunige http://orcid.org/0000-0002-2241-3194
Charlotte Dieteren http://orcid.org/0000-0002-7809-1657
Judith Bom http://orcid.org/0000-0003-4477-7594
Maarten Engel http://orcid.org/0000-0002-1774-7042
Bruno Meessen http://orcid.org/0000-0002-0359-8621
Igna Bonfrer http://orcid.org/0000-0002-8570-0393

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
