## [Reviewer comments · BMJ Open]

ARTICLE DETAILS

TITLE (PROVISIONAL)	Health seeking behaviours and private sector delivery of care for non-communicable diseases in low- and middle-income countries: protocol for a systematic review
AUTHORS	Brindley, Callum; Wijemunige, Nilmini; Dieteren, Charlotte; Bom, Judith; Engel, Maarten; Meessen, Bruno; Bonfrer, IEJ

VERSION 1 – REVIEW

REVIEWER	Hirschhorn, Lisa Northwestern University Feinberg School of Medicine, medical Social Sciences
REVIEW RETURNED	26-Sep-2022

GENERAL COMMENTS	The authors describe a protocol for a systematic review of NCD care seeking in the private sector in LMICs. The review addresses an important and often ignored components of the care system and one which is increasingly used for management of NCDs in these countries In the introduction-they state that “. Importantly, NCDs have an etiology linked to behavioral risk” while this is correct for some, there are other factors including environmental and genetic. It is important to recognize as strategies for prevention and control need to address multiple etiologies There are a number of statements, which while correct, that need references such as the lack of inclusion of private sector in national NCD planning. Methods The aims are not quite as focused as the title and the introduction. “What are the health care seeking behaviors of individuals with regard to NCDs in LMICs? a. From whom is health care obtained? b. What are the determinants that influence the choice of health care? c. What are the individual and collective consequences of these health care choices? 2. What are the major issues, knowledge gaps, opportunities and challenges that arise from private providers supplying health care services for NCDs in LMICs? The authors should consider focusing on private sector versus broader access if I am reading these aims correctly. The aims also include importantly private sector provider factors, but this is also not as clearly stated in the title and introduction or synthesis. It is
---

	also unclear in the aims and introduction if this is all levels of healthcare (prevention, promotion, curative at the PHC level and hospital level)-although the levels are included in the search (see also comment about synthesis) There is good use of PRISMA and PICOT, although not clear if this is only adults, and there is mention of informal providers but not seen in search terms. This should be clarified In the determinants-interested if other population and community level factors. In the analysis, there is note of a framework-is that the described organization or does this draw from any health system/utilization frameworks (ex. High quality health systems). The grouping while helpful, I think miss some of the aims and are a little confusing. For example, In describing determinants, I was unsure how health outcomes was not a consequence? I was also interested in why location (urban/rural) and level of care (for example, MI care versus PHC-based HTN) was captured and included in the synthesis as these reflect very different scope and levels of care The consequences are important but care will need to be taken if only descriptive versus comparisons to be able to determine causality of use of private (versus other) sector if that is part of the goals.
--	--

REVIEWER	Arakelyan, Stella The University of Edinburgh Usher Institute of Population Health Sciences and Informatics
REVIEW RETURNED	09-Jan-2023

GENERAL COMMENTS	Thanks for inviting me to peer review the review protocol "Health seeking behaviours and private sector delivery of care for non-communicable diseases in low- middle-income income countries: protocol for a systematic review". I have a few comments for the authors to consider which I detail below: Abstract (1) I would like the authors to be more specific in their descriptions of the review objectives. (2) The authors need to provide justifications for language and publication date restrictions. Also, please add to the Abstract that you will include English literature only. (3) What has informed the selection of the databases? Will the authors consider Epistemonikos, Scopus, PsycInfo, ASSIA? (4) Will the authors consider the use of electronic support tools (e.g., Covidence) to minimize bias in screening, quality assessment and data extraction? If yes, please add. (5) How the results will be synthesised, especially quantitative evidence? This should be made clear in the Abstract. Introduction (6) What is the authors' definition of the private sector? This needs to be clearly stated in the introduction as well. (7) The authors should give a better indication of the existing evidence base on this topic, including any other systematic reviews. (8) Did the authors run preliminary searches in PROSPERO, JBI Evidence Synthesis and other electronic platforms to ensure that there are no other current or in-progress systematic reviews on this topic? Please expand on this in the protocol.
--

	Methods The authors should be more specific in their descriptions of selection criteria. (9) Page 5, line 47 - What is the definition of adults? (10) Page 5 - What are the main determinants of interest? Would the authors consider the wider socio-economic and political context (e.g., conflict-affected?) as well? Please expand on this. (11) Page 5, line 60 - What do the authors mean by “consequences of health seeking behaviour”, please spell this out. If the authors also consider health outcomes as consequences, please list all the health outcomes of interest. (12) Need to provide justifications for a language restriction. (13) Given this is a comprehensive systematic review, I wonder if the authors will consider using database-specific filters to identify the experimental and observational studies of interest. For instance https://www.sign.ac.uk/what-we-do/methodology/search-filters/ (14) Please revise the subsection ‘Stakeholder consultation’ planned for October 2022, as it is January 2023 just now. (15) Page 9, lines 45-46 - The authors describe “health outcomes” as determinants of health-seeking behaviour, however, these can also be consequences. This reiterates the importance of providing clear and precise definitions of key concepts used in the review. Please explain. (16) How the authors will synthesise evidence extracted from randomised and non-randomised studies. Please explain. Supplementary materials (17) PRISMA checklist should state the location where the item is reported, not if the item is reported. Please revise accordingly.
--	--

VERSION 1 – AUTHOR RESPONSE

Reviewer 1

1. The authors describe a protocol for a systematic review of NCD care seeking in the private sector in LMICs. The review addresses an important and often ignored components of the care system and one which is increasingly used for management of NCDs in these countries. In the introduction-they state that “. Importantly, NCDs have an etiology linked to behavioral risk” while this is correct for some, there are other factors including environmental and genetic. It is important to recognize as strategies for prevention and control need to address multiple etiologies. There are a number of statements, which while correct, that need references such as the lack of inclusion of private sector in national NCD planning.

Response. We thank the reviewer for confirming the relevance of this systematic review and for drawing our attention to the statements in the introduction that need qualification and additional references, which we have duly added.

“the determinants and outcomes of private health care utilisation in LMICs remains insufficiently understood. Importantly, NCDs have multiple etiologies (including behavioural, environmental, and genetic) that require an organisation of services and delivery of care that is different from that of acute illnesses.[12]” p.3 in main document with tracked changes shown in red

“...Adapting health service delivery models to respond to NCDs remains a challenge.[14] In many LMICs, national NCD strategies tend to focus on the public sector to the exclusion of the private sector despite the two being entwined elements of the whole health system.[15–17] Many LMICs with universal health coverage (UHC) benefits packages also only offer limited

coverage for NCDs, such that patients must pay out-of-pocket even in public health facilities.”[18] p.3

Added references

12. WHO (2020) WHO package of essential noncommunicable (PEN) disease interventions for primary health care, <https://apps.who.int/iris/handle/10665/334186>

15. Rani et al. (2012) A qualitative study of governance of evolving response to non-communicable diseases in low-and middle- income countries, <https://doi.org/10.1186/1471-2458-12-877>

16. Nyaaba et al. (2017) Tracing Africa’s progress towards implementing the Non-Communicable Diseases Global action plan 2013-2020, <https://doi.org/10.1186/s12889-017-4199-6>

17. Boudreaux et al. (2020) Noncommunicable Disease (NCD) strategic plans in low- and lower-middle income Sub-Saharan Africa, <https://doi.org/10.1080/16549716.2020.1805165>

18. Jan et al. (2018) Action to address the household economic burden of non-communicable diseases, [https://doi.org/10.1016/S0140-6736\(18\)30323-4](https://doi.org/10.1016/S0140-6736(18)30323-4)

2. The aims are not quite as focused as the title and the introduction. The authors should consider focusing on private sector versus broader access if I am reading these aims correctly. The aims also include importantly private sector provider factors, but this is also not as clearly stated in the title and introduction or synthesis. It is also unclear in the aims and introduction if this is all levels of healthcare (prevention, promotion, curative at the PHC level and hospital level)-although the levels are included in the search (see also comment about synthesis).

Response. We agree and have clarified the review’s objectives by adding to the introduction and aligning the research questions. We describe the levels and type of care in Table 1 and in sub-section ‘Inclusion/exclusion criteria’. We have also added references to the existing conceptual frameworks on which we draw.

“This study aims to synthesize the scattered evidence on how people with NCDs choose their healthcare providers in LMICs, and the outcomes of these choices, with a focus on the understudied private sector. In particular, we concentrate on the contextual and individual factors that influence provider choice, patterns of utilisation, quality of care and financial protection. We use the notion of the patient journey and draw on the the conceptual framework of facilitators and barriers to accessing healthcare.[21–24]” p.3

“This systematic review on the determinants and outcomes of private health care utilisation is built around the questions – What factors influence private health care seeking behaviour for individuals with NCDs in LMICs? From whom is health care obtained, what are the patterns of utilization, what are the determinants that influence the use of private as opposed to public health care, and what are the outcomes of this health care seeking? This review allows us to determine the size and nature of the current literature and to identify major knowledge gaps that relate to mixed health care systems with entwined public and private providers of NCD care in LMICs.” p.4

Added references

21. Andersen (1995) Revisiting the Behavioral Model and Access to Medical Care, <https://doi.org/10.2307/2137284>

22. Manderson and Smith-Morris (2010) Introduction in Chronicity and the Experience of Illness. <https://doi.org/10.36019/9780813549736-002>

23. Levesque et al. (2013) Patient-centred access to health care: conceptualising access at the interface of health systems and populations, <https://doi.org/10.1186/1475-9276-12-18>

24. Brathwaite et al. (2020) The Long and Winding Road: A Systematic Literature Review Conceptualising Pathways for Hypertension Care and Control in Low- and Middle-Income Countries, <https://doi.org/10.34172/ijhpm.2020.105>

3. There is good use of PRISMA and PICOT, although not clear if this is only adults, and there is mention of informal providers but not seen in search terms. This should be clarified

Response. We have specified in Table 1 that the population of interest are adults aged 18 years or older and we have listed the NCDs on which we focus. We have adopted a broad definition of private provider and exclude studies at the title-abstract screening and full-text review stages rather than filter for provider types and risk missing studies. See also response to Reviewer 2, comment 6. We have also clarified the levels of care and year on which the World Bank income groups are based.

Table 1 (p.5)

Population	Adults aged 18 years or older and households at risk, or diagnosed with at least one of the following non-communicable diseases: cardiovascular diseases, diabetes, chronic respiratory diseases or cancers (tracheal, bronchus and lung; colon and rectum; pancreas; stomach; breast and prostate)
Interventions	Private sector provision of health care services all levels (e.g. primary health clinic to hospital) that involves the prevention, diagnosis, treatment and/or management of non-communicable disease
Control	Not applicable
Outcomes	Determinants and outcomes of health care seeking behaviour, at the individual and population level
Timeframe	1 January 2010 to 30 June 2022
Setting	Low-income and middle-income countries following the World Bank 2022 classification [33]

4. In the analysis, there is note of a framework-is that the described organization or does this draw from any health system/utilization frameworks (ex. High quality health systems). The grouping while helpful, I think miss some of the aims and are a little confusing. For example, In describing determinants, I was unsure how health outcomes was not a consequence? I was also interested in why location (urban/rural) and level of care (for example, MI care versus PHC-based HTN) was captured and included in the synthesis as these reflect very different scope and levels of care. The consequences are important but care will need to be taken if only descriptive versus comparisons to be able to determine causality of use of private (versus other) sector if that is part of the goals.

Response. We acknowledge that our conceptual framework was not sufficiently clear, and we have addressed this by clarifying the review’s aims and adding references to the existing frameworks on which we draw. See response to comment 2. We have also added a reference to the Donabedian model which helps conceptualise quality of care. We will outline our conceptual framework in more detail in the actual review rather than the protocol. See also response to Reviewer 2, comment 15.

“We will extract the determinants of selecting a health care provider, grouped into three categories: 1) individual factors such as demographics and health; 2) contextual factors such availability and accessibility of health care; and 3) perception of providers such as quality and competence. Lastly, we extract outcomes of selecting a health care provider, which we will group into three categories: 1) patterns of utilization including characteristics of patients and equity issues; 2) quality of care including the Donabedian model [40] domains of process and outcomes as well as patient satisfaction and empowerment; and 3) spending and financial protection as captured by indicators such as catastrophic health spending and impoverishment.” p.8

Added references

40. Donadebian (1980) The definition of quality and approaches to its assessment, ISBN: 9780914904489

We appreciate the ambiguity around the way we used “health outcomes” as a determinant and reframed it as perception of providers. See also response to Reviewer 2, comment 10.

The reason we differentiate by location and level of care is because we anticipate that the studies we find will have a wide scope and different levels of care which cannot be directly compared, as you mention. However, we would like to keep the study broad by including a range of locations (urban/rural) and levels of care. It would be important to extract these factors so they can be described when analysing determinants and outcomes presented in different studies, and taken into consideration if any comparisons are made.

Reviewer 2

1. I would like the authors to be more specific in their descriptions of the review objectives.

Response. We agree and have clarified the review’s objectives by adding to the introduction and aligning the research questions. We have also added references to the existing conceptual frameworks on which we draw. See also response to Reviewer 1, comment 2.

“This study aims to synthesize the scattered evidence on how people with NCDs choose their healthcare providers in LMICs, and the outcomes of these choices, with a focus on the understudied private sector. In particular, we concentrate on the contextual and individual factors that influence provider choice, patterns of utilisation, quality of care and financial protection. We use the notion of the patient journey and draw on existing conceptual frameworks of health seeking behaviour and the facilitators and barriers to accessing healthcare.[21–24]” p.3

“This systematic review on the determinants and outcomes of private health care utilisation is built around the questions – What factors influence private health care seeking behaviour for individuals with NCDs in LMICs? From whom is health care obtained, what are the patterns of utilization, what are the determinants that influence the use of private as opposed to public health care, and what are the outcomes of this health care seeking? This review allows us to determine the size and nature of the current literature and to identify major knowledge gaps that relate to mixed health care systems with entwined public and private providers of NCD care in LMICs.” p.4

Added references

21. Andersen (1995) Revisiting the Behavioral Model and Access to Medical Care, <https://doi.org/10.2307/2137284>

22. Manderson and Smith-Morris (2010) Introduction in Chronicity and the Experience of Illness. <https://doi.org/10.36019/9780813549736-002>

23. Levesque et al. (2013) Patient-centred access to health care: conceptualising access at the interface of health systems and populations, <https://doi.org/10.1186/1475-9276-12-18>

24. Brathwaite et al. (2020) The Long and Winding Road: A Systematic Literature Review Conceptualising Pathways for Hypertension Care and Control in Low- and Middle-Income Countries, <https://doi.org/10.34172/ijhpm.2020.105>

2. The authors need to provide justifications for language and publication date restrictions. Also, please add to the Abstract that you will include English literature only.

Response. We agree and have noted in the abstract that the review includes English literature only and acknowledge this as a limitation. We have added further explanation for our publication date restrictions in the Methods section.

“We focus on the timeframe from January 2010 to June 2022, which is defined by large increases in the NCD disease burden in LMICs and the first High-Level Meeting of the UN

General Assembly on the Prevention and Control of Non-communicable Diseases.[29,31] This also avoids duplication of an earlier systematic review into the private provision of health services in LMICs conducted in 2011.[32]” p.5

“Strengths and limitations of this study

Although we will consider all LMICs, this systematic review is limited to studies published in English, so does not include relevant literature in other languages.

We limit our attention to literature from the past decade which we contend is most relevant given recent changes in disease burden, risk factors and demographics in LMICs.” p.2

3. What has informed the selection of the databases? Will the authors consider Epistemonikos, Scopus, PsycInfo, ASSIA?

Response. We have added a reference to Bramer et al. 2017 which helped us to develop the rationale for using our selection of databases, while working with the expert from the Erasmus Medical Centre library. Including even more databases than we currently do would indeed widen the number of studies that we find. However, given our focus on cardiovascular diseases, cancers, respiratory diseases and diabetes, the database PsycInfo is likely to return studies outside the scope of our review and both Scopus and ASSIA have important overlap with the databases we selected. We thank the reviewer for the suggestion to look into Epistemonikos; we were not aware of this database and will use it for the introduction of our review where we will summarise previous systematic reviews.

4. Will the authors consider the use of electronic support tools (e.g., Covidence) to minimize bias in screening, quality assessment and data extraction? If yes, please add.

Response. We agree with the importance of minimizing bias in screening, quality assessment and data extraction. We considered several software tools and have now added the references that informed our decision.

“The articles resulting from the search will be screened by a team of two pairs (CB-JB & NW-CD) working in the Rayyan Reference Manager.[38,39]” p.8

Added references

38. Van der Mierden et al. (2019) Software tools for literature screening in systematic reviews in biomedical research, <https://doi.org/10.14573/altex.1902131>

39. Harrison (2020) Software tools to support title and abstract screening for systematic reviews in healthcare, <https://doi.org/10.1186/s12874-020-0897-3>

For example, Van der Mierden et al. 2019 found that Rayyan was the best free tool and scored similarly to Covidence in terms of functionality. They write, “The only mandatory feature not supported by this tool (Rayyan) is distinct title/abstract, and full text phases. This can be circumvented by exporting all results after the title/abstract phase and importing them for the full-text phase.” We plan to do this and we have opted for Rayyan because of its functionality and contribution to making research including software more accessible to all (i.e. open science).

5. How the results will be synthesised, especially quantitative evidence? This should be made clear in the Abstract

Response. While space is of course limited in the abstract, we have now added more detail on this and clarified that the main analysis will be a descriptive synthesis and what results we will focus on. We will not perform a meta-analysis of quantitative data, which we acknowledge as a limitation and appears as a box with the abstract in BMJ Open.

“The team will discuss and analyse the data extracted, which we will consolidate using a descriptive synthesis including a summary of the evidence, gaps, and limitations. As described above in the data extraction, we will synthesise the results into three categories of determinants of selecting a health care provider (individual factors, contextual factors and perception of providers) and three categories of outcomes of selecting a health care provider_(patterns of utilization, quality of care, and spending and financial protection). We will also develop a

synthesis of the results that concern the interaction of health seeking determinants and outcomes. A flow diagram of the inclusion/exclusion pathways and the descriptive statistics of the included studies and their outcomes will be developed to complement the analysis. We will not perform a meta-analysis because we expect highly diverse study characteristics, including design type, setting, intervention and outcome.” p.9

“Strengths and limitations of this study

We will not perform a meta-analysis because we expect highly diverse study characteristics, including design type, setting, intervention and outcome” p.2

6. What is the authors’ definition of the private sector? This needs to be clearly stated in the introduction as well.

Response. We have a definition of the private sector in the sub-section ‘Inclusion/exclusion criteria’.

“We use the World Health Organization’s operational definition of the private health sector as the individuals and organizations that are neither owned nor directly controlled by governments, and are involved in the provision of health services (i.e. formal and informal providers as well as for-profit and non-profit entities.[30]” p.5

7. The authors should give a better indication of the existing evidence base on this topic, including any other systematic reviews.

Response. We agree that the existing evidence base is important and include the rational for this review with the support of references in the introduction. We will discuss in dept the existing evidence in the actual review as opposed to the protocol given limited space and the emphasis here on methods. See also our responses to your comments 3 and 8.

8. Did the authors run preliminary searches in PROSPERO, JBI Evidence Synthesis and other electronic platforms to ensure that there are no other current or in-progress systematic reviews on this topic? Please expand on this in the protocol.

Response. We can confirm that we did run a preliminary search and appreciate the reminder to duly add this the protocol.

“This study is guided by the Preferred Reporting Items for Systematic Reviews and Meta-Analyses (PRISMA) and its extension for literature searches PRISMA-S.[27,28] PRISMA is a systematic approach to map and synthesise existing evidence, identify knowledge gaps and inform future research. This review was registered with PROSPERO (CRD42022340059) on 15 June 2022. We checked that there were no current or in-progress systematic reviews on the same topic by searching PROSPERO, the Research Registry, and the Open Science Framework.” p.4

9. The authors should be more specific in their descriptions of selection criteria. What is the definition of adults?

Response. We have added a clearer definition of selection criteria, including on adults, to our sub-section ‘Inclusion/exclusion criteria’ and Table 1. See also response to Reviewer 1, comment 3.

“We focus on adults aged 18 years of older...” p.5

Table 1 (p.5)

Population	Adults aged 18 years or older and households at risk, or diagnosed with at least one of the following non-communicable diseases: cardiovascular diseases, diabetes, chronic respiratory diseases or cancers (tracheal, bronchus and lung; colon and rectum; pancreas; stomach; breast and prostate)
------------	---

Interventions	Private sector provision of health care services all levels (e.g. primary health clinic to hospital) that involves the prevention, diagnosis, treatment and/or management of non-communicable disease
Control	Not applicable
Outcomes	Determinants and outcomes of health care seeking behaviour, at the individual and population level
Timeframe	1 January 2010 to 30 June 2022
Setting	Low-income and middle-income countries following the World Bank 2022 classification [29]

10. What are the main determinants of interest? Would the authors consider the wider socio-economic and political context (e.g., conflict-affected?) as well? Please expand on this.

Response. We appreciate the opportunity to make this clearer and we have provided more detail on the main determinants of interest under the subsection 'Data extraction' (see also response to your comments 11, 15 and 16). This complements Table 2, which contains search elements and key terms.

"We will pilot and refine a data extraction framework including **study details** such as authors, title and year; and **study characteristics** such as research design, sample size, studied country, disease and provider type. We will also extract the **determinants of selecting a health care provider**, grouped into three categories: 1) *individual factors* such as demographics and health status; 2) *contextual factors* such availability and accessibility of health care; and 3) *perception of providers* such as quality and competence. Lastly, we extract **outcomes of selecting a health care provider**, which we will group into three categories: 1) *patterns of utilization* including characteristics of patients and equity issues; 2) *quality of care* including the Donabedian model[40] domains of process and outcomes as well as patient satisfaction and empowerment; and 3) *spending and financial protection* as captured by indicators such as catastrophic health spending and impoverishment." p.8

Added reference

40. Donabedian (1980) The definition of quality and approaches to its assessment, ISBN: 9780914904489

We will consider contextual factors to the extent they influence the selection of a health care provider (e.g. availability, convenience, affordability, accessibility, appropriateness of services). We will exclude irregular contextual circumstances (e.g., conflict, economic or political crises) because these are not generalizable (see sub-section 'Inclusion/exclusion criteria' p.5)

11. What do the authors mean by "consequences of health seeking behaviour", please spell this out. If the authors also consider health outcomes as consequences, please list all the health outcomes of interest.

Response. We have changed this wording to outcome throughout the protocol and made the review's research questions clearer (see responses to comments 1 and 10). Table 2 provides a list of health outcomes of interest. We will use broad terms to include the largest set of outcomes given the expected heterogenousness of the studies focusing on different NCD and health care settings. We will exclude studies at the title-abstract screening and full-text review stages rather than apply narrow filters on the search.

Search element	Key terms
----------------	-----------

What: noncommunicable diseases	noncommunicable diseases; cardiovascular disease; heart attack; stroke; myocardial infarction; hypertension; diabetes; cancer (top 5 for each sex: lung, breast, colorectum, prostate, stomach, liver, cervix, thyroid); chronic respiratory disease; chronic obstructive pulmonary disease; asthma)
Where: health care providers in low- and middle-income countries	hospital, health clinic, community health centre, health centre, health post, family practitioner, general practitioner, primary care, secondary care, tertiary care, specialist, home care, nursing home, pharmacy, public provider, private provider, nonprofit provider, faith-based provider, low-income countries, middle-income countries, lower-middle income countries, upper-middle income countries, list of all LMIC countries
Why and what happens: determinants and outcomes of health care decisions / health seeking behaviour	quality; cost; affordability; accessibility; proximity; distance; opening hours; in-network provider; reimbursement; reputation; health service delivery; health outcomes; utilization; out-of-pocket payment; indirect costs; catastrophic health care expenditure; debt; health inequalities; health information; supply; fee for service; empowerment; patient reported experience

12. Need to provide justifications for a language restriction.

Response. We considered including all of the WHO languages (English, French, Spanish, Russian and Chinese) but do not have two fluent speakers for each language to independently screen and review the articles. We believe that translation software would do an imperfect job and have acknowledged this language restriction as a limitation. See also response to your comment 2 above.

13. Given this is a comprehensive systematic review, I wonder if the authors will consider using database-specific filters to identify the experimental and observational studies of interest. For instance: SIGN search filters

Response. We thank you for the helpful suggestion and have considered database-specific filters, which would make the search more refined and return fewer results. However, every filter that is added has the risk of losing relevant studies since filters cannot capture 100% of the studies with a particular research design. As such, we prefer to search broadly for all study types with exclusion filters only for editorials and case reports. We will filter design type at the title-abstract and full-text review stages, which will take more time but ensure completeness.

14. Please revise the subsection ‘Stakeholder consultation’ planned for October 2022, as it is January 2023 just now.

Response. Thank you for pointing this out. We have dropped this subsection and noted our intentions elsewhere under the subsection ‘Ethics and dissemination’.

“The results will be published in a topic relevant journal and presented at related scientific events attended by policy makers, academics and health care practitioners from LMICs, and later published in a topic relevant journal.” p.9

15. The authors describe “health outcomes” as determinants of health-seeking behaviour, however, these can also be consequences. This reiterates the importance of providing clear and precise definitions of key concepts used in the review. Please explain.

Response. We agree with this point and view health care seeking from a provider as potentially a reiterative process with feedback loops (e.g. effectiveness of treatment and patient satisfaction influences future health seeking behaviour). We have made this clearer in the review’s aims, research questions and data extraction. We have also added a reference to the Donabedian model which helps conceptualise quality of care as both a determinant and outcome. See also responses to comments 1 and 10 above.

16. How will the authors synthesise evidence extracted from randomised and non-randomised studies. Please explain.

Response. We have added more detail and clarified that the main analysis will be a descriptive synthesis and outlined the results that we will focus on. See response to comment 5 above. In the sub-section ‘Quality assessment’, we explain how we will use the Mixed Methods Appraisal Tool to assess methodological quality. This tool includes questions specific to different design types. For example, whether confounders are accounted for in the design and analysis of non-randomised quantitative studies. Our assessment using MMAT will inform our synthesis of the randomised and non-randomised designs (e.g. highlighting the findings of higher quality studies).

17. PRISMA checklist should state the location where the item is reported, not if the item is reported. Please revise accordingly.

Response. We have edited the checklist to include the relevant page references. See uploaded Supplementary Material.

VERSION 2 – REVIEW

REVIEWER	Hirschhorn, Lisa Northwestern University Feinberg School of Medicine, medical Social Sciences
REVIEW RETURNED	14-May-2023

GENERAL COMMENTS	The authors have chosen to conduct a systematic review to fill an important gap in the knowledge needed to address the gap in capacity and delivery fo services to prevent, diagnose and manage the growing burden of NCDs in LMICs focusing on the often ignored role of the private sector. The methods are clearly described and appropriate. There were a few areas where more detail are needed . the authors should be more clear if and how they will focus on private sector. In addition, it would be helpful to understand if they were planning to use any health system and/or quality frameworks to understand barriers and facilitators for access, as well as the quality and equity of delivery of NCDs. In addition, it would be good to understand why mental health as a major cause of morbidity and contributing to mortality and with an enormous gap in capacity and delivery of care in LMICs. It would also be helpful to include a potential limitations.
---

REVIEWER	Arakelyan, Stella
-----------------	-------------------

	The University of Edinburgh Usher Institute of Population Health Sciences and Informatics
REVIEW RETURNED	15-May-2023

GENERAL COMMENTS	Thanks to the authors of this systematic review protocol for addressing most of my comments. Although the authors have revised their work and provided clarifications, I still have residual methodological concerns: (a) Searches in databases – records retrieved from systemically searching Scopus, ASSIA and other key databases will almost always overlap, hence the reason for de-duplication and further staged screening. Please provide a stronger justification for not considering earlier proposed databases in your search strategy. (b) Outcomes - it is still unclear to the reader what the outcomes of interest are. (c) Determinants - socio-cultural, economic and political contextual factors are important determinants of health-seeking in LMICs. For instance, due to prolonged political instability, poor political governance and economic crises, trust in public health institutions may erode shaping future health-seeking behaviour. Socio-cultural beliefs, personal, but also social networks' past and current experiences of accessing care are drivers of future health seeking. Please explain why these contextual factors are not of interest to your work.
--

VERSION 2 – AUTHOR RESPONSE

We are grateful to the reviewers for their careful reading of our revised paper and their valuable comments. We have responded below to each of the reviewers' points with new text added in red. We have also developed a diagram of our conceptual framework based on your helpful advice.

Reviewer 1

The authors have chosen to conduct a systematic review to fill an important gap in the knowledge needed to address the gap in capacity and delivery of services to prevent, diagnose and manage the growing burden of NCDs in LMICs focusing on the often ignored role of the private sector. The methods are clearly described and appropriate. There were a few areas where more detail are needed. The authors should be more clear if and how they will focus on private sector. In addition, it would be helpful to understand if they were planning to use any health system and/or quality frameworks to understand barriers and facilitators for access, as well as the quality and equity of delivery of NCDs. In addition, it would be good to understand why mental health as a major cause of morbidity and contributing to mortality and with an enormous gap in capacity and delivery of care in LMICs. It would also be helpful

to include a potential limitations

Response. We thank the reviewer for reiterating the relevance of this systematic review and highlighting a few points that could be clearer.

Concerning our focus on the private sector, we explain in the introduction that we intend to synthesize evidence on how people with NCDs choose private health care providers in LMICs and the outcomes of

these choices. In the exclusion/criteria section and appendix, we reference our use of the WHO's definition of the private health sector and how we will search for relevant articles. We have now made our focus on the private sector more explicit; the Introduction now contains the following: "This study aims to synthesize the scattered evidence on how people with NCDs choose their healthcare providers

in LMICs, and the outcomes of these choices, in the understudied private sector. This will provide important new insights into the determinants and outcomes of private health care utilisation for NCDs in LMICs."

We agree that a conceptual framework would help the reader understand our approach so we have developed a diagram describing our underlying framework and added references to the theoretical frameworks on which we draw: "We draw on existing theoretical approaches to health seeking behaviour and the facilitators and barriers to accessing healthcare.[21–25] Figure 1 depicts our conceptual framework of the non-linear interaction of the determinants and outcomes along the patient journey of seeking private health care for NCDs."p.5

References

21. Andersen RM. Revisiting the Behavioral Model and Access to Medical Care: Does it Matter? *J Health*

Soc Behav 1995;36:1. doi:10.2307/2137284

22. Manderson L, Smith-Morris C. Introduction: Chronicity and the Experience of Illness. In:

Introduction: Chronicity and the Experience of Illness. Rutgers University Press 2010. 1–18.

doi:10.36019/9780813549736-002

23. Levesque J-F, Harris MF, Russell G. Patient-centred access to health care: conceptualising access at

the interface of health systems and populations. *Int J Equity Health* 2013;12:18. doi:10.1186/1475-9276-12-18

24. Brathwaite R, Hutchinson E, McKee M, et al. The Long and Winding Road: A Systematic Literature

Review Conceptualising Pathways for Hypertension Care and Control in Low- and Middle-Income Countries. *Int J Health Policy Manag* 2020;:1. doi:10.34172/ijhpm.2020.105

25. Donabedian A. The definition of quality and approaches to its assessment. Ann Arbor, Mich: Health

Administration Press 1980.

We also agree that mental health is a serious and understudied cause of morbidity and mortality.

However, we think this condition deserves its own systematic review and this systematic review's scope

is restricted to the four major groups of NCDs responsible for 80% of all premature deaths:

cardiovascular diseases, diabetes, chronic respiratory diseases and cancers.

3

Lastly, we agree that it's important to specify potential limitations and we've done this in the format required by BMJ Open (a list of strengths and limitations after the abstract that will be formatted as a separate textbox).

Reviewer 2

Thanks to the authors of this systematic review protocol for addressing most of my comments.

Although the authors have revised their work and provided clarifications, I still have residual methodological concerns:

(a) Searches in databases – records retrieved from systemically searching Scopus, ASSIA and other key

databases will almost always overlap, hence the reason for de-duplication and further staged screening.

Please provide a stronger justification for not considering earlier proposed databases in your search strategy.

(b) Outcomes - it is still unclear to the reader what the outcomes of interest are.

(c) Determinants - socio-cultural, economic and political contextual factors are important determinants of health-seeking in LMICs. For instance, due to prolonged political instability, poor political governance

and economic crises, trust in public health institutions may erode shaping future health-seeking

behaviour. Socio-cultural beliefs, personal, but also social networks' past and current experiences of

accessing care are drivers of future health seeking. Please explain why these contextual factors are not

of interest to your work.

Response. We thank the reviewer for reading our revised paper and drawing our attention to some remaining points that need clarification.

(a) We appreciate the reiteration of this point and have taken the time to again discuss the databases in depth with the librarian on our team. The databases we have proposed cover close to 40,000 journals

and include the core combination of Embase, MEDLINE, Web of Science, and Google Scholar, which Bramer et al. 2017 show consistently achieve a recall of 95% of relevant articles. As the reviewer can hopefully appreciate, there are some decisions in preparing a search strategy and we have decided to follow the expert on our team while we fully acknowledge that other experts like yourself have different

views of which databases are most fitting. The results of our discussion are as follows. We do not include

ASSIA since it covers about 500 journals many of which are already indexed in the major databases we

use. 85% of the ASSIA journals are also from the UK and the US, which is unlikely to help us increase the coverage of articles based on LMIC settings. We do not include SCOPUS because it does not have

standardised indexing terms (i.e. emtree), nor support long search strategies. Furthermore, SCOPUS has considerable overlap with Embase which is also published by Elsevier and included in our search strategy. As shared in our response to the previous round of peer review, we will use Epistemonikos as

per your suggestion to identify relevant systematic literature reviews. We will only include original research in the systematic review itself, but we will cite previous reviews in the introduction to summarise existing evidence and gaps.

(b and c) Regarding the outcomes and determinants of selecting a private health care provider, we agree that more detail could help the reader and we have developed a diagram representing our underlying conceptual framework and have added this as Figure 1 to our manuscript. It includes examples of determinants and outcomes of interest that are also more comprehensively listed in the search terms. We also agree that socio-cultural, economic and political contextual factors are important

determinants of health-seeking in LMICs and we have added this helpful wording to the section of data

extraction.

4

“We draw on existing theoretical approaches to health seeking behaviour and the facilitators and barriers to accessing healthcare.[21–25] Figure 1 depicts our conceptual framework of the non-linear interaction of the determinants and outcomes along the patient journey of seeking private health care for NCDs.”p.5

“We will extract the determinants of selecting a health care provider, grouped into three categories: 1) individual factors such as demographics and health; 2) contextual factors such availability and accessibility of health care; and 3) perception of providers such as quality and competence. Lastly, we extract outcomes of selecting a health care provider, which we will group into three categories: 1) patterns of utilization including characteristics of patients and equity issues; 2) quality of care including

the Donabedian model[25] domains of process and outcomes as well as patient satisfaction and empowerment; and 3) spending and financial protection as captured by indicators such as catastrophic

health spending and impoverishment. We will also consider socio-cultural, economic and political contextual factors, which are important determinants of health-seeking in LMICs.”p.9